# Emergency Medicine Perspectives: The Importance of Bystanders and Their Impact on On-Site Resuscitation Measures and Immediate Outcomes of Out-of-Hospital Cardiac Arrest

**DOI:** 10.3390/jcm12216815

**Published:** 2023-10-28

**Authors:** Kamil Bednarz, Krzysztof Goniewicz, Ahmed M. Al-Wathinani, Mariusz Goniewicz

**Affiliations:** 1Department of Emergency Medicine, Medical University of Lublin, 20-081 Lublin, Poland; kamil.bednarz@umlub.pl; 2Department of Security, Polish Air Force University, 08-521 Deblin, Poland; 3Department of Emergency Medical Services, Prince Sultan bin Abdulaziz College for Emergency Medical Services, King Saud University, Riyadh 11451, Saudi Arabia; ahmalotaibi@ksu.edu.sa

**Keywords:** cardiopulmonary resuscitation, out-of-hospital cardiac arrests, Emergency Medical Teams, ROSC, VT/VF rhythms, witness presence, amiodarone administration, bystander interventions, survival rates, public training

## Abstract

Introduction: Out-of-hospital cardiac arrests (OHCAs) represent critical medical emergencies in which timely interventions can make a significant difference in patient outcomes. Despite their importance, the role of on-scene witnesses during such events remains relatively unexplored. Aim of the Study: This research seeks to shed light on the influence of witnesses, especially family members, during OHCAs and the effect of their interventions, or the absence thereof, on outcomes. Drawing from existing literature, our working hypothesis suggests that the presence of a witness, particularly one who is knowledgeable about CPR, can increase the likelihood of obtaining the return of spontaneous circulation (ROSC), potentially enhancing overall survival rates. Methods: Using a retrospective analytical method, we thoroughly reviewed medical records from the Lublin Voivodeship between 2014–2017. Out of 5111 events identified using ICD-10 diagnosis codes and ICD-9 medical procedure codes, 4361 cases specifically related to sudden cardiac arrest were chosen. Concurrently, 750 events were excluded based on predefined criteria. Results: Both basic and advanced EMS teams showed higher rates of CPR initiation and an increased likelihood of obtaining ROSC. Notably, the presence of a trained EMS professional as a witness significantly increased the chances of CPR initiation. The presenting rhythms most often detected were ventricular tachycardia (VT) and ventricular fibrillation (VF). Different urgency codes were directly linked to varying ROSC outcomes. When witnesses, especially family members, began chest compressions, the use of amiodarone was notably higher. A significant finding was that 46.85% of OHCA patients died without witnesses, while family members were present in 23.87% of cases. Actions taken by witnesses, especially chest compressions, generally extended the overall duration of patient care. Conclusion: The crucial influence of witnesses, particularly family members, on OHCA outcomes is evident. Therefore, it is essential to increase public awareness of CPR techniques and rapid intervention strategies to improve outcomes in emergency situations.

## 1. Introduction

Sudden cardiac arrest (SCA) stands as one of the most critical and pressing global health emergencies of our time [1,2]. In Europe alone, estimates suggest that SCA affects between 67 to 170 out of every 100,000 residents each year, making it a paramount cause of death on the continent [3]. This alarming number, however, only provides a glimpse into a worldwide challenge. Depending on geographic regions and their varying healthcare practices, there are stark contrasts in the incidence and outcomes of SCA. For example, Asia experiences a concerning hospital discharge survival rate of just 2% [4], while North America [5], Europe [6], and Australia [7] showcase slightly more promising figures, with survival rates of 11%, 9%, and 12%, respectively [3].

The disparity in these survival rates is not merely coincidental but is influenced by a multitude of factors. Healthcare infrastructure, the availability and quality of emergency medical services, public awareness campaigns, and the extent of CPR training available to the general population have been identified as key determinants [8]. Regions with advanced emergency response systems and widespread public CPR training tend to report higher survival rates [9,10], whereas those with limited resources or where the public is less informed about CPR often face more challenging outcomes.

The International Liaison Committee on Resuscitation (ILCOR), established in 1992, has been instrumental in enhancing global resuscitation practices [11]. By annually reviewing medical publications and synthesizing findings into actionable guidelines, ILCOR emphasizes the importance of bystander-initiated CPR as a critical factor in improving survival rates [11]. These efforts by ILCOR have inspired real-world changes. There are numerous instances in which trained and informed bystanders have been pivotal in saving lives during SCA events, emphasizing the importance of such interventions [12].

Our study aims to determine the impact of bystander-initiated CPR on the outcomes of out-of-hospital cardiac arrests and to identify factors influencing the initiation of CPR across different socio-cultural contexts.

## 2. Materials and Methods

### 2.1. Study Design and Data Source

This study was a retrospective analysis of medical records from the Lublin Voivodeship Emergency Medical Service (EMS), with a focus on the years 2014–2017. We primarily sourced our data from call-out cards and medical rescue activity cards, which the Emergency Medical Teams (EMTs) completed at the scene. This documentation provides details regarding patient data, medical interventions, and outcomes. 

### 2.2. Study Regions and Population

Our research focused on the Lublin Voivodeship, which is equipped with a robust emergency medical system designed to address cardiac emergencies, among other health crises. Within the region, the Emergency Medical Service (EMS) operates through a tiered response system that swiftly prioritizes cardiac arrests, ensuring rapid deployment of resources to optimize patient outcomes.

Our research encompassed medical documentation from both Specialist and Basic Emergency Medical Teams (EMTs). Specifically, 7 Specialist EMTs, equipped with advanced life support (ALS) capabilities, and 14 Basic EMTs, equipped with basic life support (BLS) capabilities, were included. These teams are strategically stationed to ensure swift response times, and they regularly undergo rigorous training to address cardiac emergencies, emphasizing timely cardiopulmonary resuscitation (CPR) and advanced cardiovascular life support (ACLS) interventions.

The districts covered in this study included Lublin City, Świdnik, Łęczna, Piaski, Kraśnik, and several other municipalities within the Lublin region. Collectively, these districts are home to a diverse population of more than 641,000 residents. The EMS in this region collaborates closely with local hospitals and cardiologists to streamline the continuum of care for cardiac arrest patients, from the scene of the emergency to hospital admission.

### 2.3. Criteria for Inclusion and Exclusion

From an initial total of 5111 events identified based on ICD-10 diagnosis codes and ICD-9 medical procedure codes, 4361 events were incorporated into the study, while 750 were excluded. The primary reasons for these exclusions encompassed a range of issues. Some records lacked a confirmed SCA based on the Medical Rescue Activity Cards. In an attempt to capture all possible SCA incidents, we evaluated procedure codes such as “introduction of an oral-pharyngeal tube”, “introduction of a nasal-pharyngeal tube”, “other mechanical ventilation”, “bronchial tree toilette”, “cardioversion”, “CPAP”, “non-invasive mechanical ventilation”, and “endotracheal intubation”. However, these procedures were occasionally performed, even in the absence of an SCA. Furthermore, we encountered instances of incorrect coding by the EMS leaders using ICD-10 codes, and situations in which two EMS teams were dispatched to a single OHCA to support resuscitation efforts. Additional complications arose from gaps in documentation and illegible cards. Collectively, these factors led to the exclusion of 14.67% of the original 5111 cards from our study.

### 2.4. Data Collection and Analysis

#### 2.4.1. Data Entry and Verification

The data extracted from the call-out cards and medical rescue activity cards were systematically entered into Microsoft Excel by two independent members of the research team. To ensure accuracy, the entered data was cross-checked and verified by a third team member, who was blinded to the data source.

#### 2.4.2. Statistical Analysis

We employed STATISTICA software version 12.5 (StatSoft Poland) for our statistical processing. Prior to analyzing the continuous variables, we tested for normality of data distribution using the Shapiro–Wilk test. For data that followed a normal distribution, we presented the mean and standard deviation (SD). For data not adhering to a normal distribution, we relied on the median and interquartile range (IQR). Qualitative data were presented as frequencies and percentages. Statistical relationships were established using Chi-square tests and Pearson’s correlation, with significance levels set at 0.05.

#### 2.4.3. Study Variables

In our examination of “witness reactions”, we captured the actions undertaken by individuals prior to the EMTs’ arrival. These actions encompassed immediate CPR administration, usage of automated external defibrillators (AEDs), passive observation, and providing assistance in crowd control. Following the Utstein Style definition, we classified bystander CPR as CPR carried out by someone not acting as part of an organized emergency response system in response to a cardiac arrest [13]. To elucidate further, EMS personnel, be they physicians, nurses, or paramedics, were deemed bystanders only if they were not part of the organized emergency response system for that specific cardiac arrest scenario.

In our examination of “witness reactions”, we documented actions taken by individuals before the EMTs’ arrival. These actions included immediate CPR administration by either trained or untrained bystanders. “CPR Initiated”, in this context, refers to the commencement of CPR, irrespective of who initiated it, before the professional EMS response.

“Duration of patient care” measured the timeframe from when the EMTs received the emergency call until they departed from the scene. This included the entire response phase—initial patient assessment, necessary interventions, and preparations for either patient transport or transitioning of care.

Lastly, we also classified “urgency” using Code 1 and Code 2, where Code 1 highlighted the most urgent situations.

#### 2.4.4. Primary and Secondary Outcomes

The core focus of our study was the rate of the return of spontaneous circulation (ROSC) observed in OHCA patients within the Lublin Voivodeship region from 2014–2017. Our primary objective was to discern the potential influence of factors such as the time taken for the first responder to arrive and the presence, as well as the type, of bystander (be it a family member, passerby, or trained first responder) on the ROSC rate. For the sake of clarity, we define the ROSC rate as the proportion of OHCA patients who, after undergoing resuscitation attempts, successfully regained spontaneous circulation, which could manifest either at the incident location, during transportation, or upon reaching a healthcare facility.

For the purpose of this study, we defined ROSC as the return of a palpable pulse or measurable blood pressure following a cardiac arrest event, without the need for chest compressions. Consistent with guidelines suggested by ILCOR, an event was recognized as ROSC if spontaneous circulation was sustained for at least 30 s without the necessity for cardiopulmonary resuscitation.

In this study, when we reference “survival rate”, we specifically mean the ROSC rate, as this is our primary measure of survival. Future research or analyses might consider other metrics of survival, such as survival upon hospital admission, survival upon discharge, or survival over a specific period.

Our secondary objectives were more comprehensive, delving into the immediate medical interventions carried out by witnesses or bystanders at the scene. We assessed whether CPR was initiated before the emergency medical services (EMS) reached the scene and explored the availability and employment of automated external defibrillators by bystanders. Further, where available, we analyzed data regarding the frequency and quality of chest compressions. Lastly, we aimed to identify occurrences in which medications were administered before EMS’s arrival, even though this scenario is less common among those who are not medically trained.

In addition to these primary and secondary outcomes, we undertook an analysis of the efficiency of the emergency response system in the Lublin region. Our focus was on determining its impact on patient outcomes, especially in cases where bystanders played a significant role during the emergency.

### 2.5. Ethical Considerations

In line with ethical standards, all extracted data underwent a pseudonymization process. As such, none of the data used contained any personal identifiers, as defined by the Act of 10 May 2018 in regards to personal data protection (Journal of Laws 2018, item 1000). This study was sanctioned by the Bioethics Committee of the Medical University in Lublin, with the decision number KE-0254/232/2018, ensuring that it met all ethical guidelines.

### 2.6. Methodological Strengths and Limitations

This study, being retrospective, comes with intrinsic strengths and limitations. It offers a comprehensive view of emergency interventions over several years. However, the retrospective design may introduce biases linked to record-keeping and reporting. We adopted rigorous selection criteria and data processing steps to minimize these potential pitfalls.

### 2.7. Rationale behind Data Selection

SCA, the focal point of this study, was chosen due to its profound health consequences and variations in survival rates across regions. The rich data from the call-out cards and the medical rescue activity cards enable an in-depth exploration of real-world emergency situations. This study thus becomes a pivotal resource for medical professionals, policymakers, and those passionate about public health.

### 2.8. Excluded Variables

In our study, we consciously chose not to delve into age and gender demographics in analyzing the outcomes of OHCAs. While these factors undoubtedly influence health outcomes, our primary objective was to focus on the immediate circumstances, responses, and interventions related to OHCAs. Incorporating age and gender would have introduced additional complexities that might detract from our primary focus. Furthermore, age and gender analyses require a considerably larger dataset to draw statistically significant conclusions, which we believed was outside the scope of our current investigation. Future studies building on our findings might consider these demographic variables to provide a more nuanced understanding of OHCA outcomes.

## 3. Results

We conducted a retrospective analysis of emergency interventions in the Lublin region from 2014 to 2017. Out of a larger dataset of 277,998 EMT interventions, 5111 events were identified using ICD-10 diagnosis codes and ICD-9 medical procedure codes related to sudden cardiac arrest. Following our inclusion criteria, we analyzed 4361 of these events in depth, while excluding 750, according to specific criteria.

### 3.1. Factors Influencing the Success Rates of CPR and ROSC

The data presented in Table 1 highlights several variables impacting the success rates of CPR initiation and the subsequent achievement of ROSC. Influencing factors include the type of EMS team, the team leader’s profession, the urgency code, bystander presence and response during cardiac arrest, and the initial ECG rhythm.

For the purpose of this study, “CPR initiated” refers to the commencement of cardiopulmonary resuscitation by anyone at the scene prior to the arrival of the responding EMS team.

EMS Team: Basic EMS teams showed more frequent resuscitation initiation (39.98%) compared to specialist teams. They also had a higher ROSC success rate of 19.61%.

Team Leader’s Profession: Paramedic-led resuscitations had a higher initiation rate (48.86%), as well as a success rate of 25.57% for achieving ROSC.

Witness Presence: The presence of witnesses during an out-of-hospital cardiac arrest (OHCA) significantly influenced CPR initiation, with rates surging to 87.91% for EMS members and 75.61% for other bystanders. ROSC achievement was notably higher with an EMS member presence, reaching a rate of 56.59%.

Bystander Response: ROSC rates increased to 46.23% when witnesses provided full cardiopulmonary resuscitation.

Initial ECG Rhythm: Cases with an initial ECG rhythm of VT/VF demonstrated a CPR initiation rate of 100% and an ROSC achievement rate of 60.23%.

Urgency Code: While the urgency code influenced ROSC outcome, its effect on CPR initiation was not significant.

In summary, our results emphasize the critical role of prompt response, effective leadership, and the significant impact of bystander intervention on CPR and ROSC outcomes.

### 3.2. Relationship between Witness Reaction and Drug Administration

Table 2 delves into the relationship between bystander interventions during a cardiac arrest and the subsequent drug choices made by EMS during CPR.

Our findings predominantly suggest that bystander interventions do not significantly influence the majority of drug selections during CPR. This is evident for several drugs, indicating a consistent drug administration approach, irrespective of bystander actions.

However, a significant exception was observed with regards to amiodarone. When bystanders promptly administered chest compressions, the use of amiodarone increased, accounting for 26.42% of administrations. This is in contrast to scenarios in which bystanders either applied different CPR techniques or did not react promptly.

For commonly used drugs like epinephrine, their administration appeared to remain consistent, regardless of witness reactions.

In essence, while most drug decisions during CPR are seemingly unaffected by bystander actions, the increased use of amiodarone after immediate chest compressions by bystanders is a noteworthy finding that requires further study.

### 3.3. Duration of Patient Care Based on Various Variables

Table 3 delves into the factors influencing the length of patient care after cardiac arrest events. One notable determinant is the initiation of resuscitation. Commencing resuscitation extends care duration considerably, averaging 58.65 min, highlighting the resource intensity and crucial nature of resuscitation for these patients.

As depicted in Table 3, care duration exhibits notable disparities, based on factors such as resuscitation onset, cardiac arrest location, and bystander response. Analysis unveils a direct link between the cardiac arrest setting and the ensuing care duration. For example, cardiac arrests at workplaces have an average care duration of 52.00 min, contrasting sharply with incidents at homes, which average a shorter 34.86 min duration, despite being more common.

Bystander reactions also significantly influence EMS care duration. The data indicate an average care period of 39.59 min when witnesses abstain from CPR. However, this duration surges to an average of 60.31 min when bystanders provide chest compressions.

In summary, the duration of post-cardiac arrest patient care is influenced by a combination of factors, notably the immediate medical intervention, cardiac arrest location, and actions of witnesses present.

### 3.4. Presence and Reactions of Witnesses during OHCA

Out-of-hospital cardiac arrests (OHCAs) frequently occur without warning and can often happen away from the watchful eyes of potential bystanders. The data depicted in this figure reveal that a significant 46.85% of these events unfold without any witnesses present (Figure 1). The prevalence of unwitnessed OHCAs underscores the challenges in ensuring timely intervention, which is paramount for improving the odds of patient survival.

A closer look at OHCAs occurring in the presence of witnesses exhibits a notable trend. Family members, who usually share a deep emotional connection with the patient, were on the scene for 23.87% of these incidents. This statistic highlights the vital role that CPR training could play for the general public, as they are often the first responders in these critical moments. Delving into the reactions of these witnesses offers a deeper understanding of the community’s preparedness to handle such emergencies (Figure 2).

A troubling trend surfaces from the data: a staggering 60% of witnesses refrained from any resuscitative action—a stark reminder of the pivotal nature of immediate interventions during cardiac events. However, there are some encouraging statistics as well: 27.79% of witnesses commenced full CPR, substantially boosting the odds of victim survival. Additionally, 11.34% of witnesses employed chest compressions. Even if they do not include a complete CPR process, these singular compressions can keep the blood circulating until professional help arrives.

## 4. Discussion

The principal objective of this research was to elucidate the role and influence of bystanders on the outcomes and subsequent emergency medical services’ intervention strategies during out-of-hospital cardiac arrests.

Research has been pivotal in underscoring the bystander’s role during out-of-hospital cardiac arrests. Studies from Riggs, Birkun, and Huang reported a commendable inclination among witnesses to provide CPR during emergencies [14]. However, a chasm exists between intent and action. Factors such as prior CPR training, understanding of its importance, and even the age demographic play a role in this gap.

A foray into the bystander psyche reveals a mix of hesitations and fears. A study by Latsios et al. provided a window into this hesitation, revealing the emotional conflicts that hold many back [15]. Dainty’s findings added depth, illustrating fears of inadequacy or of causing inadvertent harm [16]. This emotional turmoil warrants a discussion on comprehensive public training that is not only technique-focused, but which also addresses the psychological barriers to initiating CPR.

The core findings from our research shed light on distinct patterns in patient care duration post-cardiac arrest. Crucially, the immediacy of resuscitative measures, the location of the cardiac incident, and the response of bystanders each profoundly influence the care duration. Notably, our results underscore the invaluable role bystanders play in shaping the trajectory of emergency responses, particularly for time-sensitive conditions such as cardiac arrests.

The observed differences in care durations based on the location of cardiac arrest, particularly the longer average durations at workplaces, might reflect the complexities inherent to handling emergencies in these environments. Possible challenges could stem from accessibility issues or the rapid procurement of essential medical equipment. The significant increase in care duration when bystander CPR was involved underscores the critical role immediate interventions, especially chest compressions, play in the overall management of cardiac arrests. Interventions initiated by bystanders may shift the clinical trajectory of cardiac arrest victims, indicating a potential need for more extended and intensive care by EMS teams.

A prolonged delay between the onset of a cardiac arrest and the initiation of resuscitation efforts and defibrillation is linked to deteriorating prospects for ROSC and favorable neurological outcomes. Early CPR and prompt defibrillation significantly enhance the survival chances of victims. As early as 1774, individuals were designated to assist drowning victims, and the benefits of such bystander interventions have been confirmed in numerous studies [17]. 

In our research, while discussing ROSC as an outcome, it is crucial to specify the time frame associated with the absence of CPR that could influence ROSC. The ILCOR suggests a duration of 30 s without CPR as a significant marker [11]. Aligning with this, our study adopted the ILCOR’s recommendation and defined ROSC based on a 30 s absence of CPR performance. It is essential to note that this standardization aids in drawing comparisons with other studies and understanding the profound effects of even brief interruptions in CPR on the survival prospects of OHCA victims.

Our research findings indicate that in the majority of cases, bystanders did not initiate resuscitation efforts. In studies conducted by Ballesteros-Pena et al., resuscitation measures were taken in 22.9% of cases, with similar results presented in Henry’s studies. In J. Sielski’s research, this figure reached 32.6%, while in the work of D. Gacha et al., it stood at 56% [18,19,20,21].

While our study did not specifically investigate the impact of the Good Samaritan laws on bystander intervention during OHCAs, it is important to acknowledge their potential influence. The Good Samaritan laws, existing in numerous jurisdictions, aim to protect individuals who assist those in peril [22]. These laws can shape the likelihood of bystander involvement. Within these legal frameworks, bystanders offering voluntary assistance during emergencies are often protected from liability, provided that their actions are in good faith and not reckless. The awareness or unfamiliarity with these laws might impact the bystander’s willingness to act due to fear of potential legal repercussions [23]. Thus, promoting understanding of these protections might enhance public confidence, fostering more proactive bystander responses [24].

Our study’s geographic scope spans districts including the large city of Lublin and smaller towns like Piaski, as well as rural areas, capturing the responses and interventions of EMTs within a diverse population setting. One pivotal aspect of our findings revolves around bystander reactions prior to EMT arrival, a crucial factor in emergency scenarios, particularly with conditions which are as time-sensitive as SCAs. Furthermore, the “duration of patient care” metric, which we will present, offers an encompassing view of the EMT response, from the initial alert to the conclusion of their intervention at the scene. As we delve into the intricacies of these results, it is essential to bear in mind the retrospective nature of our methodology and the profound implications our findings may hold for the broader medical community and public health initiatives

Although our research did not focus on the influence of socioeconomic factors on bystander CPR during OHCAs, it is noteworthy that other studies have pointed to disparities in this intervention based on the socioeconomic status of the area in which the cardiac arrest occurs [25,26]. Such disparities suggest that communities with limited resources might be at a disadvantage when it comes to public health initiatives, CPR training opportunities, and accessibility to medical facilities. This aspect was not the central focus of our investigation, but future research in this domain might benefit from a deeper understanding of these dynamics, ensuring that interventions are not only effective but also equitable.

Another challenge is the misidentification of OHCA cases. Atypical presentations can mislead even the most willing bystander, causing delays or hesitations in response. Our study substantiated this observation, drawing a correlation between unwitnessed OHCAs and decreased ROSC rates. The increased ROSC rates among non-family witnesses suggest factors such as emotional detachment or perhaps the sheer number of bystanders in public places, leading to a higher likelihood of intervention [27].

Comparing our findings with those of other studies in the domain reveals certain discrepancies. The variations, particularly regarding bystander responses, could be attributed to differences in the demographic profiles, the pervasiveness of CPR training initiatives, or public awareness campaigns in the respective study areas. Moreover, cultural factors, regional medical protocols, and community trust in medical establishments might also play a part. Our study, rooted in its specific geographic and demographic context, naturally presents unique insights, and while comparisons with other research offer valuable perspectives, inherent regional and methodological differences can explain some of the disparities observed.

Our data provide insight into the comprehensive approach to CPR, emphasizing both chest compressions and rescue breaths. We found an upward trajectory in ROSC outcomes, aligning with prior assertions that proactive witness interventions significantly impact survival rates [28].

Our findings indicate a marked increase in amiodarone administration when cardiac arrest was witnessed and immediate chest compressions were initiated. While our study highlights this unique trend, the underlying reasons remain speculative. Could it be that EMS personnel perceive these incidents—in which bystanders act promptly—as scenarios with a higher chance of survival, thus favoring the early administration of a potent antiarrhythmic? Or is it perhaps a reflection of an unconscious bias influenced by the bystander’s proactive approach? This observation underscores the need for further studies regarding EMS decision-making dynamics in the context of bystander interventions.

The immediate reaction of bystanders could have ramifications beyond just CPR initiation. As our data suggests for the amiodarone administration pattern, EMS decisions might be subtly influenced by the nature of bystander involvement. This introduces a new dimension to our understanding of pre-hospital care dynamics. It poses the question: Are there other unseen ways in which the first link in the chain of survival—the bystander—impacts subsequent medical interventions and, by extension, patient outcomes?

Our study also highlights the significance of amiodarone. Amiodarone is an antiarrhythmic drug crucial for treating the life-threatening arrhythmias often seen in sudden cardiac arrests. Its effectiveness is heightened when administered early during resuscitation, potentially augmenting the chances of achieving the return of spontaneous circulation (ROSC). Given that immediate chest compressions by witnesses can offer an earlier window for EMS intervention, it is plausible that EMS personnel are more inclined to use amiodarone in these cases, capitalizing on its benefits when introduced early in the resuscitation process.

In comparison to those in the existing literature, our findings illuminate a less-explored dimension—the potential influence of immediate bystander reactions on subsequent EMS decisions. While several studies have rightfully emphasized the importance of bystander CPR in regards to survival rates, our research goes a step farther by suggesting that these early interventions might subtly shape subsequent EMS strategies, especially those regarding drug administration choices, such as the use of amiodarone.

While public training predominantly emphasizes CPR techniques, it is essential for bystanders to recognize the importance of early interventions in influencing subsequent medical treatments, such as drug administration by medical professionals [29]. Although the general public is not expected to administer drugs or engage in advanced life support (ALS), a deeper understanding of the broader implications of timely bystander CPR can highlight its critical role in the entire emergency response process. A comprehensive training program that offers insights into the significance of early intervention and its potential downstream effects, without delving into the specifics of drug choices, could foster a more informed and proactive approach to bystander interventions [30].

While bystander intervention during OHCAs is not a novel topic, our research uniquely delves into the intricate interplay between the bystander’s immediate response and its subsequent implications on EMS interventions. This inquiry is not only relevant but essential in a field that constantly seeks to improve the survival rates of OHCA victims. Our study seeks to address a specific gap in the understanding of how bystander intervention dynamics influence, directly or indirectly, the choice and effectiveness of subsequent medical interventions.

Our findings, when placed in the broader landscape of global health, underscore the urgent need for tailored public health strategies. The intricate nuances and variables brought to light in our research can act as a compass for policymakers. Recognizing the critical role of bystanders in improving OHCA outcomes, there is a compelling case to be made for the potential of universal CPR training [31]. Introducing such training early, for example, within school curricula, or making it mandatory for certain professions, could pave the way for more timely and effective interventions in cases of OHCAs. As we refine our understanding of OHCAs through research like ours, it becomes increasingly evident that a proactive, informed approach is our best weapon in combatting this global health concern.

Additionally, we cannot overlook the role of technological advancements in addressing OHCAs. The integration of AI-driven solutions in the form of wearable devices, for instance, could provide real-time feedback to bystanders performing CPR, or even alert nearby medical professionals or trained individuals to rush to the scene [32]. This convergence of technology and medical intervention could redefine the OHCA landscape in the coming years [33,34,35].

While advanced technological solutions are crucial, ensuring easy accessibility to life-saving equipment is equally paramount.

Accessibility to AEDs significantly enhances the chances of survival during an OHCA. Immediate defibrillation can increase the likelihood of successful resuscitation, especially in ventricular fibrillation cases. Hence, public access defibrillation (PAD) programs, which aim to improve AED accessibility in public places, can be game changers. The public’s awareness and competence in using these devices, coupled with strategies ensuring their widespread availability, might bridge the gap between early intervention by bystanders and the arrival of medical professionals.

## 5. Limitations

Our study, while offering valuable insights, is not without its limitations. One primary concern is the geographical scope of the data collection, which predominantly encompasses a specific area. This concentration potentially restricts the generalizability of our findings to other populations or settings, given the potential for cultural, socio-economic, or regional variations in bystander reactions to OHCA.

Another notable limitation of our study is the absence of age and gender-based analysis in our findings. The rationale for this omission stems from the primary objective of our research, which centered around the broader environmental, emotional, and medical factors affecting bystander responses. While age and gender undoubtedly play roles in health outcomes and responses, we determined that integrating these dimensions could divert our primary focus. Nevertheless, we acknowledge the significance of these factors and emphasize the need for future research to incorporate and delve into age and gender-specific influences on OHCA outcomes.

Furthermore, we acknowledge the potential for recall bias, particularly if witnesses were interviewed after the fact. Recollections of such an emotionally charged event might not be entirely accurate or consistent across respondents. This could be compounded by the so-called “observer effect”, wherein the mere consciousness of being observed or studied can subtly change a person’s behavior. In the context of our research, this might mean witnesses either consciously or unconsciously exaggerated or underplayed their actions or reactions during the cardiac arrest event.

The reliability of the data remains a perennial challenge in such studies. By relying on reported actions and conditions, we recognize the possibility that the data might not always capture the nuances or full spectrum of reality on the ground. Reactions, emotions, or reasons for not performing CPR could be either underreported or misinterpreted. Moreover, our dive into the underlying medical conditions associated with OHCA, while comprehensive, is limited by the quality and consistency of medical history documentation. There is the potential that some conditions, especially for those without a rich medical history or from marginalized communities, might have been overlooked.

One significant limitation of our research pertains to the omission of post-resuscitation outcomes, specifically the lack of detailed data for factors such as hospital admission rates, neurological outcomes upon hospital discharge, and long-term impacts experienced by survivors. The exclusion of these outcomes from our study means that we might not have captured a comprehensive picture of the post-event health trajectory and the quality of life of survivors. Given the importance of such outcomes in gauging the effectiveness of interventions and in shaping post-resuscitation care strategies, we acknowledge this as a significant gap in our findings.

Our discussion of the emotional reasons for witnesses not initiating CPR is not exhaustive. A more in-depth psychological evaluation might offer more detailed insights into the vast array of factors influencing bystander behavior. Moreover, while we have highlighted certain challenges, like the delayed recognition of OHCA by dispatchers due to symptoms such as seizure-like movements, we have not extensively explored the efficacy of dispatcher training in recognizing and guiding CPR during OHCAs.

Lastly, it is worth noting that there might be other confounding variables that were not accounted for in our study. Aspects such as the speed of EMS arrival, the availability and usage of AEDs, or the general public’s awareness and education on OHCAs in the area could play pivotal roles in the outcomes we observed.

While our study provides a crucial step forward in understanding bystander responses to OHCA, these limitations underline the necessity for ongoing, diversified research in this area.

## 6. Conclusions

Our research underscores the critical importance of bystander-initiated CPR in enhancing the outcomes of out-of-hospital cardiac arrests (OHCAs). Our findings suggest a pronounced disparity in survival outcomes, depending on the presence or absence of timely bystander intervention. We advocate for a community-driven approach, emphasizing the need for widespread public training and strategic placement of AEDs, ensuring that every OHCA victim receives an optimal chance of survival.

## Figures and Tables

**Figure 1 jcm-12-06815-f001:**
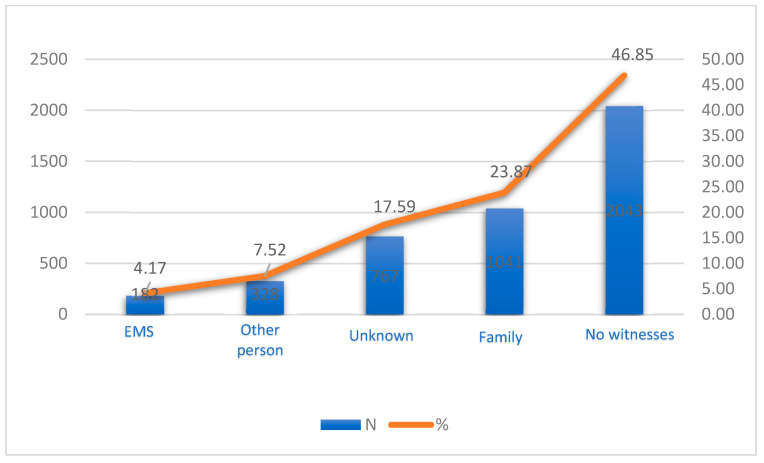
Witnesses of OHCA.

**Figure 2 jcm-12-06815-f002:**
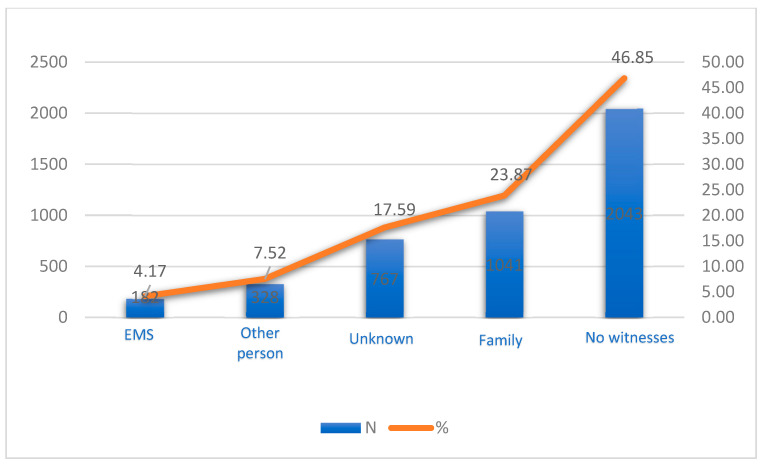
Witnesses’ reactions to OHCA.

**Table 1 jcm-12-06815-t001:** Comparative analysis of factors influencing CPR initiation and achievement of ROSC.

Factors/Variables	CPR Initiated		ROSC Achieved	
	Yes	No	Yes	No
Type of team—n (%)				
Specialist	1030 (29.87)	2418 (70.13)	393 (11.40)	3055 (88.60)
Basic	365 (39.98)	548 (60.02)	179 (19.61)	734 (80.39)
	Chi2 = 33.8854; *p* = 0.0000		Chi2 = 42.6732; *p* = 0.0000	
Team leader—n (%)				
Physician	1043 (28.84)	2574 (71.16)	393 (10.87)	3224 (89.13)
Paramedic	214 (48.86)	224 (51.14)	112 (25.57)	326 (74.43)
Nurse	138 (45.10)	168 (54.90)	67 (21.90)	239 (78.10)
	Chi2 = 97.9915; *p* = 0.0000		Chi2 = 96.3949; *p* = 0.0000	
Urgency code—n (%)				
Code 1	1133 (31.46)	2468 (68.54)	443 (16.97)	3158 (83.03)
Code 2	262 (34.47)	498 (65.53)	129 (12.30)	631 (87.70)
	Chi2 = 2.6138; *p* = 0.1059		Chi2 = 12.0178; *p* = 0.0005	
Witnesses of OHCA—n (%)				
No witnesses	103 (5.04)	1940 (94.96)	14 (0.69)	2029 (99.31)
Family	537 (51.59)	504 (48.41)	191 (18.35)	850 (81.65)
Other person	248 (75.61)	80 (24.39)	120 (36.59)	208 (63.41)
EMS	160 (87.91)	22 (12.09)	103 (56.59)	79 (43.41)
Unknown	347 (45.24)	420 (54.76)	144 (18.77)	623 (81.23)
	Chi2 = 1476.0690; *p* = 0.0000		Chi2 = 784.0014; *p* = 0.0000	
Witnesses’ Response—n (%)				
No CPR	402 (43.27)	527 (56.73)	161 (17.33)	768 (82.67)
Full CPR + AED	384 (90.57)	40 (9.43)	196 (46.23)	228 (53.77)
Chest compressions	159 (91.91)	14 (8.09)	59 (34.10)	114 (65.90)
No data	450 (15.87)	2385 (84.13)	156 (5.50)	2679 (94.50)
	Chi2 = 1347.0320; *p* = 0.0000		Chi2 = 633.4416; *p* = 0.0000	
Initial ECG Rhythm—n (%)				
Asystole/PEA	1017 (26.09)	2881 (73.91)	341 (8.75)	3557 (91.25)
VF/VT	259 (100.00)	0 (0.00)	156 (60.23)	103 (39.77)
Other	16 (76.19)	5 (23.81)	12 (57.14)	9 (42.86)
Unknown	103 (56.28)	80 (43.72)	63 (34.43)	120 (65.57)
	Chi2 = 681.5126; *p* = 0.0000		Chi2 = 678.4285; *p* = 0.0000	

Legend and footnotes: Abbreviations used in the table include CPR for cardiopulmonary resuscitation, ROSC for return of spontaneous circulation, OHCA for out-of-hospital cardiac arrest, EMS for Emergency Medical Services, AED for automated external defibrillator, ECG for electrocardiogram, PEA for pulseless electrical activity, VF for ventricular fibrillation, and VT for ventricular tachycardia. “CPR Initiated” refers to the start of cardiopulmonary resuscitation by anyone (including bystanders) at the scene prior to the arrival of the official EMS team. All data is presented as a number (n) and percentage (%). Chi-square (Chi2) values and their associated *p*-values are given for each category, indicating significant associations.

**Table 2 jcm-12-06815-t002:** Witness reactions and EMS drug use frequency.

Drugs	Witnesses’ Reaction			
	No CPR	Full CPR	Chest Compressions	No Data
Epinephrine—n (%)				
Yes	364 (90.55)	338 (88.02)	149 (93.71)	318 (87.36)
No	38 (9.45)	46 (11.98)	10 (6.29)	46 (12.64)
	Chi2 = 5.9815 *p* = 0.1125			
Amiodarone—n (%)				
Yes	84 (20.90)	66 (17.19)	42 (26.42)	55 (15.11)
No	318 (79.10)	318 (82.81)	117 (73.58)	309 (84.89)
	Chi2 = 11.0618 *p* = 0.0114			
Atropine—n (%)				
Yes	147 (36.57)	132 (34.38)	61 (38.36)	133 (36.54)
No	255 (63.43)	252 (65.63)	98 (61.64)	231 (63.46)
	Chi2 = 0.9162 *p* = 0.8215			
Sodium Bicarbonate—n (%)				
Yes	101 (25.12)	67 (17.45)	39 (24.53)	85 (23.35)
No	301 (74.88)	317 (82.55)	120 (75.47)	279 (76.65)
	Chi2 = 7.7544 *p* = 0.0514			
Dopamine—n (%)				
Yes	91 (22.64)	78 (20.31)	35 (22.01)	80 (21.98)
No	311 (77.36)	306 (79.69)	124 (77.99)	284 (78.02)
	Chi2 = 0.6685 *p* = 0.8806			
Fluid Therapy—n (%)				
Yes	264 (65.67)	242 (63.02)	111 (69.81)	214 (58.79)
No	138 (34.33)	142 (36.98)	48 (30.19)	150 (41.21)
	Chi^2^ = 7.0693 *p* = 0.0697			

Legend and footnotes: In the table, CPR stands for cardiopulmonary resuscitation. Drug interventions are presented as a number (n) and percentage (%). The Chi-square (Chi2) values, along with their associated *p*-values, are given for each drug intervention, signifying significant associations. The columns under “Witnesses’ Reaction” indicate how witnesses responded during the event.

**Table 3 jcm-12-06815-t003:** Care duration based on resuscitation, OHCA location, and witness reaction.

Variables	Patient Care Duration				
	Mean (M)	SD	Q1	Median (M)	Q3
Initiation of Resuscitation					
Yes	58.65	23.23	44.00	57.00	71.00
No	24.86	15.23	15.00	21.00	31.00
	Z = 42.2924 *p* = 0.0000				
Location					
Home (I)	34.86	23.90	17.00	27.00	48.00
Other (II)	40.26	23.64	23.00	36.50	52.00
Public place (III)	39.29	24.26	20.00	36.00	55.00
Workplace (IV)	52.00	28.69	30.00	48.00	71.00
Care Home (V)	35.09	20.91	17.50	31.00	47.50
	H = 46.7118 *p* = 0.0000 I–II; I–III; I–IV; II–IV; III–IV; IV–V				
Witnesses’ Reaction					
No Data (I)	39.77	26.01	19.00	35.00	56.00
No CPR (II)	39.59	24.81	19.00	33.00	57.00
Full CPR +AED (III)	56.37	25.51	40.00	55.00	69.00
Chest Compressions (IV)	60.31	25.20	45.00	56.00	71.00
	H = 216.5119 *p* = 0.0000 I–III; I–IV; II–III; II–IV				

Legend and Footnotes: This table presents variables in relation to the duration of patient care, measured in minutes. The metrics used include the mean (M), standard deviation (SD), 1st quartile (Q1), median (M), and 3rd quartile (Q3). The Mann–Whitney U test was used to evaluate the data for the “Initiation of Resuscitation” category, and the Kruskal–Wallis test (H) was applied to assess the “Location” and “Witnesses’ Reaction” categories. Specific pairwise comparisons are denoted by Roman numerals. All results are significant at *p* = 0.0000, with the Roman numerals indicating the groups compared in the pairwise analyses.

## Data Availability

The datasets used and/or analyzed during the current study are available from the corresponding author on reasonable request.

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
