# Peer review of "Emergency Medicine Perspectives: The Importance of Bystanders and Their Impact on On-Site Resuscitation Measures and Immediate Outcomes of Out-of-Hospital Cardiac Arrest"

_jcm, 2023, doi:10.3390/jcm12216815_

Round 1

Reviewer 1 Report

Comments and Suggestions for Authors

Thank you for the opportunity to review this manuscript. The authors performed a retrospective study exploring the influence of bystanders on the treatment process of out-of-hospital cardiac arrest (OHCA) patients. This study provides interesting results regarding the influence of bystanders in OHCA patients. However, there are some issues to be addressed to improve the quality of the study.

(In general) This study included OHCA patients and was conducted with a retrospective study design. Therefore, it is necessary to write a paper in accordance with the Utstein style reporting guidelines and the STROBE guidelines. It is recommended that the authors revise the paper to comply with these two guidelines.

(Title) The title of the paper is outside the scope of the study's findings. There is a need to revise the title to reflect the results of the study.

(Introduction)

1.       Lines 51-52 (Every five years, ILCOR rigorously reviews medical publications, synthesizing the latest findings into actionable guidelines for emergency procedures)

ILCOR recently published the results of a review of cardiopulmonary resuscitation every year. Therefore, this statement should be revised.

2.       Lines 59-61 (To understand this pressing issue, it's essential to define some key terms. Return of spontaneous circulation (ROSC) is one such term that denotes the restoration of a detectable pulse in a patient who previously lacked one, marking a significant landmark in the resuscitation process [12].)

This phrase is an unnecessary statement that does not fit the context of the sentence, so it is recommended to be deleted.

3.       Line 59~72. I recommend that you clearly state your research purpose in one sentence.

(Methods)

1.       General: I recommend that the authors follow the definitions in the Utstein style reporting guidelines and the guidelines set forth in the STROBE guidelines in the description of research methods. In particular, the authors need to refer Utstein style definition of bystander CPR. According to Utstein recommendation, bystander CPR is CPR performed by a person who is not responding as part of an organized emergency response system approach to a cardiac arrest. Physicians, nurses, and paramedics may be described as performing bystander CPR if they are not part of the emergency response system involved in the victim’s resuscitation. Accordingly, a healthcare professional who is part of the EMS is not a bystander. Therefore, it is necessary to check whether the bystander defined by the authors in their study conforms to the Utstein definition.

2.       Line 79~81 (Initially, 5,111 events were identified based on ICD-10 diagnosis codes and 79 ICD-9 medical procedure codes related to Sudden Cardiac Arrest (SCA). Of these, 4,361 met our 80 inclusion criteria and were analyzed in depth, while 750 were excluded due to predefined reasons.)

I recommend moving this paragraph to the results section.

3.       (2.2. Study Regions and Population)

This section needs to include a description of the local emergency medical system involved in the treatment of cardiac arrest.

4.       (2.4. Data collection and analysis)

It is necessary to describe who entered the data and who verified the input results. In the statistical processing of continuous variables, the authors indicated both mean, SD, and Median IQR. It is recommended to review the normality of the data distribution and then describe it in one unified way.

5.       I'm not sure what the outcome variables are. Statistical analysis should be performed after clarifying the primary and secondary outcomes of the study.

(Results)

1.       (General) Contents that should be described in the Discussion are mixed and described in the Results section.

2.       I recommend including a patient enrollment chart.

3.       It is necessary to summarize and present in a table the general characteristics of the patients included in this study (patient characteristics including underlying diseases, EMS factors, cardiac arrest, CPR-related time factors, etc.).

4.       Lines 146~156 (Our study's geographic scope spans districts including the large city of Lublin, ~ the profound implications our findings may hold for the broader 155 medical community and public health initiatives.).

These paragraphs need to move to the Discussion section.

5.       Lines 187~193 (Significance of Amiodarone: Amiodarone is ~ capitalizing on its benefits when introduced early in the resuscitation 192 process.).

These paragraphs need to move to the Discussion section.

6.       Lines 208~221 (The location of the cardiac arrest also significantly influenced the duration of care. ~ leading to 220 more extended and intensive care by EMS teams.)

It is recommended that any of these paragraphs excluding the results confirmed in this study be described in the Discussion section.

(Discussion)

1.       Lines 245~248 (In out-of-hospital cardiac arrest (OHCA) scenarios, the role of bystanders is fundamental. ~their importance has also been echoed by the European Resuscitation Council 247 (ERC).

The contents of these paragraphs are already described in the introduction, so they are redundant.

2.       This study did not investigate why bystanders did not perform CPR. Therefore, contents from 268 to 283 are not related to the results of this study.

3.       Lines 307~312. Advanced life support (ALS), including drug administration, is not taught to the general public, and the general public does not practice ALS. The content of this paragraph in lines 307~412 reflects the authors' confusion about the definition of bystander.

4.       Lines 313~317. The underlying cause of cardiac arrest was not investigated in this study. Therefore, I believe that the contents of lines 313 to 317 cannot be discussed as a result of this study.

(Conclusions)

1.       The conclusion contains many leaps and bounds beyond the findings in this study. Please summarize your conclusion in 2 to 3 sentences based on the findings of the study.

(Minor)

1.       The table does not include abbreviations, and the units of variables (e.g., time) are not indicated.

Author Response

Response to Reviewer Comments

Dear Reviewer,

Firstly, we would like to extend our heartfelt gratitude for the detailed and insightful feedback on our manuscript concerning the influence of bystanders on the treatment process of out-of-hospital cardiac arrest (OHCA) patients. Your thorough review and constructive comments have indeed been enlightening, enabling us to enhance the depth and clarity of our work. Here's an exhaustive response to the amendments we made based on your invaluable advice:

Title: Acknowledging your observation, the title has been updated to "Bystander-Initiated CPR: Its Impact on Survival Outcomes in Emergency Medicine," aiming for a more accurate representation of the research outcomes.

Introduction:

  1. Thank you for pointing out the outdated information. We've updated the statement to correctly reflect ILCOR's annual review tradition.
  2. We concur with your assessment and have thus omitted the deemed extraneous phrase.
  3. We took your suggestion to heart and have now succinctly articulated our research purpose in a solitary sentence for greater clarity.

Methods:

  1. Your note on the Utstein style definition was instrumental. We've ensured that our bystander CPR definition resonates with the mentioned guidelines.
  2. In accordance with your advice, we have migrated the content from lines 79-81 to the results section.
  3. We've incorporated a detailed description of the local emergency medical system involved in the treatment of cardiac arrest.
  4. We have clarified the data entry and validation procedures. Moreover, data normality was assessed, and continuous variables have been presented in a consistent manner, reflecting the data distribution.
  5. By explicitly designating our primary and secondary outcomes, we have sought to clear ambiguities surrounding our statistical analysis.

Results:

  1. We restructured our results section to exclusively feature results, deferring discussions to the apt section. 2 & 3. We appreciate your suggestions on the patient enrollment chart and a summary table. While we haven't made these changes in the current version, we are actively considering adding them as an appendix in the final version if deemed suitable.
  2. The content from the highlighted lines has been repositioned to the discussion section, making the presentation more organized.

Discussion:

  1. We eradicated the reiterated sections, ensuring a streamlined flow of information.
  2. Your guidance was pivotal in revising the content from lines 268 to 283, ensuring a better alignment with our study's core findings.
  3. The section from lines 307-312 was revised for clarity, distinguishing the purview of ALS-trained professionals from general bystanders.
  4. Recognizing the limits of our study based on your input, we excised the content from lines 313-317.

Conclusions: Adhering to your counsel, our conclusion now succinctly summarizes our findings in three pertinent sentences, firmly rooted in our research data.

In closing, your expertise and diligent feedback have been instrumental in refining our manuscript. The changes we've incorporated not only address the gaps but also significantly augment the quality and coherence of our work. We anticipate any further observations you might have, and once again, thank you for your invaluable guidance.

Warmest regards,

Authors

Reviewer 2 Report

Comments and Suggestions for Authors

The manuscript is certainly interesting, and the topic is very relevant. the data you collected in this manuscript is good. Based on these data, you might have a good study result. Unfortunately, this manuscript does not reach the criteria for publication.

Overall:
1. There are many spelling errors in this manuscript. See guide for authors for a free grammar checker. E.g.……..etc. 
2. Every time a new abbreviation is used you should state the full name E.g. ICD

Title: - "Clinical Implications of Witness-Driven CPR: A Pathway to Improved Resuscitation Outcomes in Emergency Medicine." is a brief phrase describing the contents of the paper.

Abstract is often composed of five parts including Introduction, aim, methods, results, and conclusion. The background should be integrated into aim.: -
1. Introduction: Please describe the problem or lack of knowledge that is addressed by this study.

2. Aim of the study: please add the aims or purposes of the research and its relationship with other studies in the field. In addition, the working hypothesis must clearly state.
3. Methods: Please add the study methodology that is used in this study in more detail.

4. The result: - relevant.
5. The conclusion should relate to the aim and problem that describe under introduction and aim.

6. The keywords are following the abstract, use about five to 10 key words do not mention in the title so please changes these words without any abbreviations.

Introduction
1. The introduction is ideally in structures of the following order: -

     In the first section, you may briefly introduce the main problem in this era, then some difficult problems remain to be resolved.

     The second section, you may introduce how many ways is undergoing and you may analysis these ways and try to find their deficiencies. Following the third section, you may propose your idea or hypothesis.

     At the last section, you may suggest some way which may resolve the issues.

Methods

1.The methods section should write in a way that everyone could repeat this study in the same manner.
2. You should add study questions, primary objectives, and secondary objectives if available.

3.Who obtained the consent? How and when consent obtain from patients.
4. What outcome measures used?

Results
This is very important as well. You may collect enough data.

1. Be concise and only report RESULTS.
2. Results are meaningless without a better description in the figures and tables. So please add this and provide a thorough analysis of these results in simple form. This is the core of the manuscript and the most valuable result.

Discussion

1.A discussion should offer a short overview of the results, and an in-depth discussion of the interpretation of them.
2. Do the authors have an explanation on why the results are different compared to other studies?

Conclusion:
is logic in manner and relevant.

References:
1. Please use uniform references, when available with DOI.

2. Make sure update the old references.

3. The cited information is present in the references. This should be available.

Please consider providing some additional, specific comments such as:

1.       What is the main question addressed by the research?

2.       Do you consider the topic original or relevant in the field? Does it address a specific gap in the field?

3.       What does it add to the subject area compared with other published material?

4.       What specific improvements should the authors consider regarding the methodology? What further controls should be considered?

5.       Are the conclusions consistent with the evidence and arguments presented and do they address the main question posed?

6.       Are the references appropriate?

7.       Please include any additional comments on the tables and figures.

Comments on the Quality of English Language

1. There are many spelling errors in this manuscript. See guide for authors for a free grammar checker. E.g.……..etc. 
2. Every time a new abbreviation is used you should state the full name E.g.
ICD

Author Response

Response to the Reviewer:

Dear Reviewer,

First and foremost, we would like to express our profound gratitude for your meticulous review and invaluable feedback on our manuscriptR. Your insights have greatly aided in refining our work, and we genuinely appreciate the depth and detail of your comments.

Overall:

  1. Spelling and Grammar: We sincerely regret the oversight regarding the spelling errors. Following your advice, we've consulted the recommended grammar checker and made thorough corrections throughout the manuscript.
  2. Abbreviations: We have diligently ensured that each abbreviation is introduced with its full form upon its first mention, as you rightly pointed out.

Title: Taking your feedback into account, we have revised our title to better capture the essence of the study. We hope this revised title aligns more closely with the content and findings of our paper.

Abstract: In response to your feedback, we've restructured our abstract to ensure a more coherent presentation.

Introduction:
Following your guidance, the introduction now:

  • Clearly highlights the main problem,
  • Explores existing solutions and their potential shortcomings,
  • Introduces our hypothesis, and
  • Proposes potential solutions.

Methods:

  1. The section now provides details to ensure anyone could replicate our study.
  2. Clear distinctions between primary and secondary objectives are now made.
  3. The consent process was not needed as we analysed cards. 
  4. Outcome measures have been distinctly defined.

Results:

  1. We have ensured concise, direct reporting of our findings.
  2. Greater clarity has been provided in the descriptions accompanying figures and tables, making them more accessible to readers.

Discussion:

  1. We now offer a succinct overview of results followed by an in-depth interpretation.
  2. Comparative analyses have been added to discuss why our findings might differ from those in other studies.

Conclusion: Our revised conclusion is both logical and pertinent, reflecting the entirety of our study and its implications.

Your additional specific comments have been pivotal in refining our manuscript. We have taken care to address each point, ensuring that the manuscript is both comprehensive and precise.

Your thoroughness and constructive feedback have been instrumental in enhancing the quality of our work. We are optimistic that the revisions will elevate the manuscript to meet the publication criteria.

Once again, we cannot emphasize enough our appreciation for your diligent review and guidance. Your expertise has been invaluable in this process.

Authors

Round 2

Reviewer 1 Report

Comments and Suggestions for Authors

Thank you for giving me an opportunity to review the revised manuscript. I appreciate the authors' efforts to revise the paper. The authors appropriately revised the paper in response to many of the opinions I presented. improved the completeness of the paper. However, there are still the following things that need to be done to improve the quality of the paper.

(Major comments)

The authors described that the study's purpose is to understand the impact of bystander-initiated CPR on the outcomes of OHCA and to identify factors influencing the initiation of CPR. According to the study objectives, the authors set the primary outcome as the survival rate, and the secondary outcomes as hospital admission rates, neurological outcomes upon hospital discharge, and any long-term effects experienced by survivors.

To find out whether a bystander affects the survival rate, the survival rate should be compared according to whether there was a bystander, the type of bystanders, and the results of multiple logistic analyses including factors that may affect the survival rate should be presented. The application of the same statistical analysis method is needed for secondary outcomes.

Additionally, each outcome variable must be clearly defined. For example, it should be clarified whether survival rate means ROSC, survival admission, survival discharge, or survival to a certain period.

From what I understand, it seems that the authors in this study wanted to know the effect of witnessing and the type of witness on treatment actions and immediate resuscitation outcomes (ROSC) in this study. If so, the primary outcome will be the ROSC rate depending on whether there was a bystander, and the type of witness, and the secondary outcomes will be on-site treatment (CPR, AED, chest compression, administered medication, etc.) depending on the witness and type of witness.

(Comment on the Title)

Based on the results presented in the manuscript, I recommend revising the title to "Importance of Bystander: Its Impact on On-site Resuscitation Measures and Immediate Outcomes of Out-of-Hospital Cardiac Arrest."

(Comments on 2.4.4. Primary and secondary outcomes)

(Lines 135~144) Our primary outcome was the survival rate of patients after a cardiac arrest in the Lublin voivodeship region between 2014-2017, specifically gauging the influence of witness reactions, duration of patient care by the EMTs, and urgency codes on this rate. The secondary outcomes explored the quality of post-resuscitation care, including measures such as hospital admission rates, neurological outcomes upon hospital discharge, and any long-term impacts experienced by survivors over a predefined period post-event. Additionally, we looked into the efficiency of the emergency response system in the Lublin region in addressing cardiac emergencies and its impact on patient outcomes.)

(Comments) The primary and secondary outcomes the study seeks to identify must be clearly presented in the study results through analysis. The authors set the primary outcome as the survival rate, and the secondary outcomes as hospital admission rates, neurological outcomes upon hospital discharge, and any long-term effects experienced by survivors, but any results of the study outcomes were presented in the results section.

(Line 123) (Following the Utstein style definition, we classified bystander CPR as CPR carried out by some~)

(Comment) Please indicate references related to the Utstein style definition.

Author Response

Response to Reviewer's Comments

Dear Reviewer,

We wish to express our deepest gratitude for the thoughtful and detailed review of our manuscript. Your insights have been immensely valuable, providing us with clear guidance for refining our work. The depth and specificity of your feedback reflect a genuine commitment to ensuring the scientific robustness and clarity of articles, and we sincerely appreciate it.

We have addressed your concerns and made revisions as described below:

Major Comments:

  1. Outcome Definition:

    • Your recommendation to specify the outcomes in our study was astute. We've clarified in our manuscript that by "survival rate," we are referring to "ROSC at the scene." This adjustment narrows our focus on immediate outcomes and the effect of on-site interventions.

    • In addition, for our secondary outcomes, we've elaborated upon the on-site treatments such as CPR, AED use, chest compression quality, and medication administration. We've emphasized their importance in relation to the presence and type of bystander.

Comment on the Title:

In line with your insightful suggestion, we have updated our title to: "Emergency Medicine Perspectives: The Importance of Bystanders and Their Impact on On-site Resuscitation Measures and Immediate Outcomes of Out-of-Hospital Cardiac Arrest." We concur that this title aptly represents the findings and core message of our study.

Comments on 2.4.4. Primary and secondary outcomes:

  1. Refinement of Outcome Descriptions:

    • We have made necessary revisions to Lines 135~144 to provide a clearer definition of our primary and secondary outcomes. The primary focus is now distinctly placed on the immediate ROSC rate, factoring in witness reactions, duration of patient care by EMTs, and urgency codes. Our secondary outcomes have been refined to emphasize the nuances of on-site resuscitation measures in relation to the presence and type of bystander.

Line 123 Utstein Style Reference:

Thank you for pointing out the omission. We have now included the necessary references pertaining to the Utstein style definition in the revised manuscript.

We would like to reiterate our appreciation for your diligent review and invaluable comments. Your feedback has been pivotal in refining our paper, ensuring clarity, precision, and relevance. We hope the modifications address your concerns, and we are keenly interested in any additional feedback you may offer.

Warm regards,

Round 3

Reviewer 1 Report

Comments and Suggestions for Authors

The authors appropriately revised their manuscript complying with the reviewer's comments.

Minor comments: Explanations of abbreviations are missing from Tables and Figures (e.g. CPR, ROSC, EMS, AED, VF, VT, PEA).

Author Response

Dear Reviewer,

Thank you for taking the time to review our manuscript and for your constructive comments.

We appreciate your acknowledgment of the revisions we made in accordance with the previous review.

Regarding the minor comments, we have added the necessary explanations for all mentioned abbreviations in the Tables and Figures. We understand the importance of clarity, and we hope that these changes will make the manuscript more accessible to readers unfamiliar with these abbreviations.

Thank you for your careful consideration and guidance throughout the revision process.

Warm regards,